# The Finnic Peoples of Russia: Genetic Structure Inferred from Genome-Wide and Y-Chromosome Data

**DOI:** 10.3390/genes15121610

**Published:** 2024-12-17

**Authors:** Anastasia Agdzhoyan, Georgy Ponomarev, Vladimir Pylev, Zhaneta Autleva (Kagazezheva), Igor Gorin, Igor Evsyukov, Elvira Pocheshkhova, Sergey Koshel, Viacheslav Kuleshov, Dmitry Adamov, Natalia Kuznetsova

**Affiliations:** 1Research Centre for Medical Genetics, 115522 Moscow, Russia; st26i900@gmail.com (G.P.); freetrust@yandex.ru (V.P.); janetka0001@bk.ru (Z.A.); goha2@mail.ru (I.G.); igor.v.evsyukov@mail.ru (I.E.); eapocheshkhova@mail.ru (E.P.); viacheslav.kuleshov@gmail.com (V.K.); nimissin@mail.ru (D.A.); 2Biobank of Northern Eurasia, 115201 Moscow, Russia; 3Department of Faculty Therapy, Faculty of Medicine, Maykop State Technological University, 385000 Maykop, Russia; 4Department of Biology with the Course of Medical Genetics, Pharmaceutical Faculty, Kuban State Medical University, Mitrofana Sedina Str., 4, 350063 Krasnodar, Russia; 5Department of Cartography and Geoinformatics, Faculty of Geography, Lomonosov Moscow State University, 119991 Moscow, Russia; skoshel@mail.ru; 6Institute for the History of Material Culture, Russian Academy of Sciences, Dvortsovaya Naberezhnaya, 18A, 191186 Saint-Petersburg, Russia; 7Dipartimento di Scienze Linguistiche e Letterature Straniere, Università Cattolica del Sacro Cuore, Largo Gemelli 1, 20123 Milan, Italy; natalia.kuznetsova@unicatt.it; 8Department of the Languages of Russia, Institute for Linguistic Studies, Russian Academy of Sciences, Tuchkov Per. 9, 199004 Saint-Petersburg, Russia

**Keywords:** Finnic people, gene pool, Y-chromosome, genome-wide SNPs, ancestral components, Karelians, Veps, Ingrian Finns, Votes, Ingrians

## Abstract

**Background:** Eastern Finnic populations, including Karelians, Veps, Votes, Ingrians, and Ingrian Finns, are a significant component of the history of Finnic populations, which have developed over ~3 kya. Yet, these groups remain understudied from a genetic point of view. **Methods:** In this work, we explore the gene pools of Karelians (Northern, Tver, Ludic, and Livvi), Veps, Ingrians, Votes, and Ingrian Finns using Y-chromosome markers (N = 357) and genome-wide autosomes (N = 67) and in comparison with selected Russians populations of the area (N = 763). The data are analyzed using statistical, bioinformatic, and cartographic methods. **Results:** The autosomal gene pool of Eastern Finnic populations can be divided into two large categories based on the results of the PCA and ADMIXTURE modeling: (a) “Karelia”: Veps, Northern, Ludic, Livvi, and Tver Karelians; (b) “Ingria”: Ingrians, Votes, Ingrian Finns. The Y-chromosomal gene pool of Baltic Finns is more diverse and is composed of four genetic components. The “Northern” component prevails in Northern Karelians and Ingrian Finns, the “Karelian” in Livvi, Ludic, and Tver Karelians, the “Ingrian-Veps” in Ingrians and Veps (a heterogeneous cluster occupying an intermediate position between the “Northern” and the “Karelian” ones), and the “Southern” in Votes. Moreover, our phylogeographic analysis has found that the Y-haplogroup N3a4-Z1927 carriers are frequent among most Eastern Finnic populations, as well as among some Northern Russian and Central Russian populations. **Conclusions:** The autosomal clustering reflects the major areal groupings of the populations in question, while the Y-chromosomal gene pool correlates with the known history of these groups. The overlap of the four Y-chromosomal patterns may reflect the eastern part of the homeland of the Proto-Finnic gene pool. The carriers of the Y-haplogroup N3a4-Z1927, frequent in the sample, had a common ancestor at ~2.4 kya, but the active spread of N3a4-Z1927 happened only at ~1.7–2 kya, during the “golden” age of the Proto-Finnic culture (the archaeological period of the “typical” Tarand graves). A heterogeneous Y-chromosomal cluster containing Ingrians, Veps, and Northern Russian populations, should be further studied.

## 1. Introduction

The modern Finnic people populate Northeastern Europe and represent the western branch of the Uralic language family [1]. Finns, Estonians, and a small ethnic group of Livonians primarily inhabit Finland, Estonia, and Latvia, respectively. Karelians, Veps, Ingrians, Votes, and Ingrian Finns inhabit Northwestern Russia, including the Republic of Karelia, the Leningrad Region, the Arkhangelsk Region, the Vologda Region, and the Tver Region. According to the 2020 Russian Population Census [2], the Finnic groups in Russia account for less than 1% (~53,000) of all Finnic speakers, yet they occupy at least one-third of the territory where the Finnic languages are spoken.

### 1.1. The Historical Background of the Finnic Peoples and Their Migrations

The history of the Finnic people is closely connected to Northeastern Europe. As a linguistic community, the Proto-Finnic groups emerged about 3 kya [3,4,5].

By the beginning of the Bronze Age, the gene pool of Northeastern Europe’s south (the Eastern Baltic region) had undergone a significant structural change. The contribution of the Mesolithic Eastern hunter–gatherers (EHG) had surpassed that of the Western hunter–gatherers (WHG). Additionally, new genetic components from the Corded Ware complex individuals had been introduced through migrations from the Pontic-Caspian steppe that drove a transition to agropastoralism [6,7,8,9] and a spread of Indo-European languages in the region [5]. The earliest genetic trace of migrations from Western Siberia inferred from autosomal (the Siberian component) and Y-chromosomal (haplogroup N3) data comes from the north of the Kola Peninsula. It dates back ~3.5 kya and is associated with the expansion of Uralic speakers [10]. Before 2 kya, this migration wave reached Scandinavia; its impact can be traced from Pre-Viking (1–749 AD) [11] to modern populations [12]. The Bronze Age populations of the Eastern Baltic region did not carry the Siberian component [9,13]. 

In the Iron Age, no later than 2.5 kya, when Finnic languages started to diversify, the Siberian autosomal component and the Y-chromosomal haplogroup N3 have been found in the ancient populations of the Tarand graves archaeological culture (Tarandgräber) on the territory of today’s Estonia [13]. The communities of the Tarand graves culture (Figure 1) had a fixed habitat and a distinct material culture, subsisted on local resources, and probably experienced a demographic rise and a significant increase in population density; a lot more Tarand graves than stone-cist graves have been discovered so far (in particular, see the map in [14]). The archaeological period of the so-called “typical” Tarand graves (I-IV c. AD), characteristic of Estonia, Northern and Western Latvia, Southwestern Finland, and Northwestern Russia during the Roman Iron Age [15] (pp. 174–177, 306–307), could be defined as the “golden age” of the Proto-Finnic civilization. Finns, Estonians, Livonians, and Votes all originated within the geographic boundaries of the Tarand graves culture. In particular, Votes began separating from Northern Proto-Estonian ancestors in the 1st c. AD and later became part of the Novgorod Republic [16,17] (p. 21). In the mid-1st millennium AD (from 7–8 c. AD to 12–14 c. AD [15] (pp. 256, 316), some of the Tarand grave culture descendants from the south of today’s Finland moved to the Karelian Isthmus, where the Korela tribes later came into being [18,19] (p. 11).

Migrations of the Finnic tribes had intensified by the 9–11th centuries AD, leading to further ethnic splits. The populations of the Ladoga Kurgan “Chud” culture from the South Ladoga regions and Beloozero “Ves” regions [17] (p. 4) gave rise to modern Veps (on Veps, see also a thorough monograph [19]). In the 11–12th centuries, their kindred tribes further dispersed eastward to the Northern Dvina basin. In the 12–17th century (especially before the 14th c.), these tribes appeared in the Russian chronicles as Zavoloch Chud [17] (pp. 111–113); by the 19th century, they had adopted Russian as their first language [17] (p. 148).

Other migrations are related to the branching of the Korela and Veps tribes. First, from the end of the 1st millennium AD (and especially from the 12th to 13th century) until no later than the 17th century, Ingrians branched off the Southern Korela tribes, who lived in the south of the Karelian Isthmus, and spread south, down by the Oredež river and along the southern coast of the Gulf of Finland and further to the Lower Luga region, no later than by the 17th century [17] (pp. 65–66), [20] (pp. 9–10), [21] (pp. 2–6). Second, the eastern part of the Korela tribes, which remained within the Old Russian State as a result of the Orekhov treaty of 1323 with Sweden, gave rise to modern Karelians [21] (pp. 170–172) (more specifically, the modern speakers of South Karelian proper dialect). Modern Livvi and Ludic Karelians in the area between the Ladoga and the Onega lakes are a result of a mix of these Karelian tribes moving southeast with the ancestors of Veps moving northeast [22] (pp. 93–94).

In the 17th century, with the T’avzino treaty of 1595 and especially the Stolbovo treaty of 1617, Sweden annexed the regions to the north and west of the Ladoga Lake, and some local groups moved inland (to the east) into the Russian Empire [23] (pp. 9–10), [24] (p. 4).

In particular, many Karelians from these areas migrated away, first, to the north and east, which gave rise to the modern speakers of Northern Karelian proper dialect, and second, to the south, which gave rise to modern “islands” of South Karelian in the Tikhvin, Valday, and Tver regions [25] (pp. 176–177).

The origin of Ingrian Finns dates back to the same period. After the Stolbovo treaty, the “Savakot” Finns of the Savo parish of Finland, an area to the north of Vyborg, and the “Äyrämöiset” Finns from the Äyräpää parish, who lived in the west of Vyborg, as well as apparently from some other Finnish parishes, more to the north, were moved by the Swedish authorities to the south. These new regions included the southern regions of the Karelian Isthmus, the south of the Neva basin, and further to the Soikkola and Kurkola peninsulas, and the scope was to re-populate this territory after the emigration of the Orthodox population [23] (pp. 11–12), [24] (pp. 4–5), [26] (pp. 469–471).

**Figure 1 genes-15-01610-f001:**
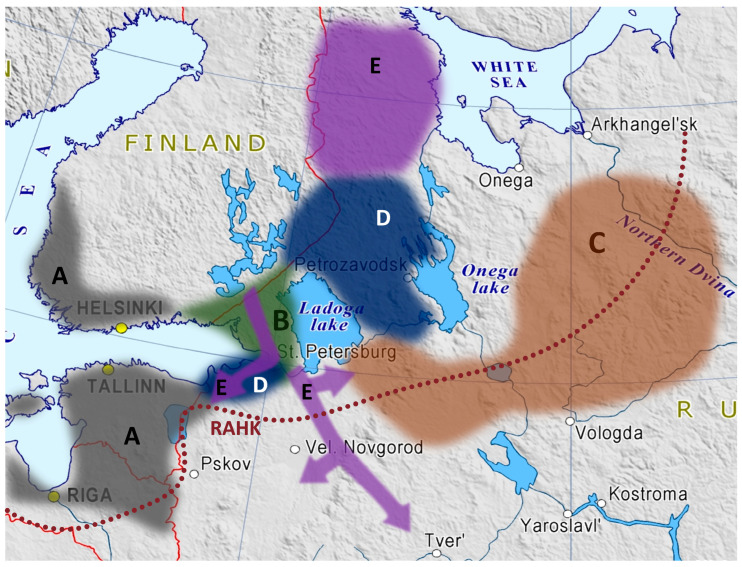
Approximate dispersal of the Eastern Finnic populations over the past 2.5 kya: A—from the 4th–3rd centuries BC to 3rd–4th centuries AD: archaeological culture of the Iron Age (“typical” Tarand graves, [15] (p. 175)); B—from the 3rd–4th to 9–11th centuries AD: the expansion of the Finnic area: the Korela tribes are being formed [19] (p. 11); C—the 9–11th centuries AD: active Finnic migrations of the tribes related to ancient Beloozero Ves’ (the alleged ancestors of modern Veps), Chud and Zavoloch Chud [17] (p. 4); D—from the end of the 1st millennium AD, but especially in the 12–15th century AD: (a) migrations of the ancestors of Ingrians from the area of the Korela tribes and the formation of Ingrians [23] (pp. 8–13), [24] (pp. 2–6); (b) Northern and Eastern migrations of Ves’ tribes and Southeastern migrations of Korela tribes and the formations of Ludic and Livvi Karelians [22] (pp. 93–94); (c) Northeastern migrations of Korela tribes after the Orekhov treaty of 1323 with Sweden and the formation of South Karelian proper [21] (pp. 170–172); E—the 17th century AD: after the T’avzino treaty of 1595 and especially the Stolbovo treaty of 1617 with Sweden, (a) mass migrations of the ancestors of Northern Karelian proper (Northern Karelian migrations), and of Tikhvin, Valdai, and Tver Karelians (Southern Karelian migrations) from the Swedish to the Russian territory [25] (pp. 176–177) and (b) the migrations of the ancestors of Ingrian Finns to the new lands of Sweden (the province of Ingria): the “Savakot” Finns from the Savo parish, the “Äyrämöiset” Finns from the Äyräpää parish, and possibly Finns from some other territories of modern Finland [26] (pp. 469–471). Dotted maroon line (“RAHK”): the southern boundary of attested Finnic toponyms [27] (p. 32).

To summarize, historically, the Finnic peoples can be subdivided into two major groups by their migratory activity: (a) a relatively sedentary group and (b) a group of active migrants. Finns, Estonians, Livonians, and Votes belong to the first group: they have lived within the geographical boundaries of their linguistic homeland for at least 2000 years. The active migration group includes today’s Karelians, as well as Ingrians, Ingrian Finns, Veps, and the medieval Finnic-speaking communities of the Russian North, which have been assimilated by the XIXth century. All these populations have been formed by migration waves from their linguistic homeland in the past 1500 years. Today, a sedentary group of Votes and most of the migratory populations (excluding the northern migrations of the Finns of Finland) live in the Russian Federation.

### 1.2. Earlier Genetic Studies on Eastern Finnic Peoples

In modern genetics, information about Finnic peoples comes mostly from large-scale studies. Detailed descriptions of Finnish and Estonian gene pools are provided in [28,29,30]. Genetic studies of Russia’s Finnic populations usually use genome-wide autosomal markers [31,32,33,34,35] and whole-genome data [36,37]), as well as individual Y-chromosome lineages [38,39,40]. Such studies have focused by now mainly on Karelians (without characterizing their subpopulations) and Veps, rarely on Ingrian Finns [34,36]; so far, there has been no published data on Votes and Ingrians. The gene pools of Karelians (N = 17), Veps (N = 12), and Ingrian Finns (N = 6) have been analyzed in a study of the dissemination of Uralic languages in Northern Europe [34]. The ancestral component typical of Uralic language speakers occurs in 11–15% of Karelians, Veps, and Ingrian Finns, which is lower than in Saami (~50%), matches the Northern Russians (13%), and is higher than in Estonians (5%). Across the entire spectrum of the ancestral components, Karelians, Veps, and Ingrian Finns have the highest affinity for the Finns of Finland.

According to whole-genome [36] and DNA [10] studies, the Uralic ancestral component occurs in modern Finnic-speaking populations at similar frequencies as the Siberian component (~10% in Karelians, N = 3, Veps, N = 4, and Ingrian Finns, N = 3). A weak similarity between Karelians (N = 35) and the populations of the Volga-Ural region has been observed in the proportion of short IBD segments (1–3 cM) in a study of ethnic gene pools covering a vast territory from the Baltic region to Lake Baikal [33]. The gene pool of Vologda Veps (N = 81) bears the highest similarity to the Northern Russians of Mezen [31]. The D-statistical analysis of whole genomes of the same populations of Veps (N = 24) and Karelians [37] has revealed the presence of the Siberian component in Karelians, Veps, Northern Russians, and the Komi people and its absence in the populations living further to the south.

Genome-wide data for Northern (N = 6) and Southern (N = 9) Karelians and the Veps of Karelia (N = 10) samples of our research team has been studied using the Affymetrix Axiom^®^ Genome-wide Human Origins 1 array (~629,000 SNPs) and reported in [35]. An in-depth analysis of their gene pools was beyond the scope of the study. Nevertheless, the study reported an ADMIXTURE modeling based on the genomic data from modern and ancient North Eurasian populations, which revealed the presence of the Siberian ancestral component in Karelians and Veps (~10%).

The Y-chromosomal gene pool of Finnic groups is characterized by an increased presence of the haplogroup N [34,39,40]. The diversity of the haplogroup is represented by its two branches: N3a4, prevalent in Karelians (20%), Veps (33%) and Finns (41%), and N3a3, more common in Estonians (28%) [39]. Of five N3a4 lineages, N3a4-B535 (Z1934), which dates back ~4 kya [40], prevails in the populations of Northeastern Europe. Its frequency reaches ~20% in Karelians, ~33% in Veps, which is slightly lower than in Finns (~44%), 20–30% in Saami, and 19–23% in the Russians of the Arkhangelsk region. There is no up-to-date data on Y-chromosome variation in Karelia’s populations (i.e., Northern, Ludic, and Livvi Karelians, as well as Veps), as well as in Ingrians, Votes, and Ingrian Finns. The Y-chromosomal gene pool of Tver Karelians has been described by the entire spectrum of haplogroups in our previous publication [41], which demonstrated their genetic affinity to the southern populations of Karelians and Veps.

In view of the above, the comprehensive analysis of the gene pools of Eastern Finnic populations for Y-chromosomal and autosomal markers with the use of a single representative sample set remains an unaccomplished task. This publication describes Karelians in most of their diversity (Northern and Tver Karelians, Livvi, and Ludic), Veps, and the poorly studied populations of Votes, Ingrians, and Ingrian Finns. These populations are analyzed in comparison with selected Russian groups of the region using a broad panel of Y-chromosome haplogroups (with a special focus on haplogroup N3a4) and a genome-wide array of over 4 million SNPs.

### 1.3. Hypotheses of This Study

Based on what is known about the earlier ethnic history and the genetics of the Finnic groups under investigation, we can expect at least the following for the results of our study:(1)A relative genetic proximity between Karelians (especially Northern Karelians), Ingrian Finns, and Ingrians because these three groups have split into several ethnicities much later than the general split of the Proto-Finnic unity happened;(2)Votes would be genetically further from all other Finnic groups under discussion because Votes have split from the Proto-Finnic unity earlier than many of the other Finnic groups and because they are the only “sedentary” Finnic group in our sample, while all other Finnic groups are “migratory”;(3)Some genetic proximity between Veps and at least some of the groups of Russian Pomors because both have been linked to the historical group of Zavoloch Chud;(4)Some genetic proximity between Votes and at least some of the groups of Novgorod Russians because Votes have been an integral part of the Novgorod Republic from very early on;(5)The southern migratory groups of Livvi, Ludic, and Tver Karelians might genetically be closer to Veps than Northern Karelians would be to Veps because these southern groups are a result of contacts between the ancestors of Karelians and Veps.

In this study, we do not consider hypotheses related to the Siberian component in the genetic pool of Eastern Finnic populations (the presence of which would be expected based on earlier research). This component will be explored in our future study on a broader panel of samples, while this study is focused on a local comparison between the Finnic groups and the neighboring Russian populations.

## 2. Materials and Methods

### 2.1. Samples

The study includes eight Finnic populations currently residing in the Russian Federation. Specifically, we studied Karelians (Northern Karelians from the Kalevalsky district, Livvi Karelians from the Olonetsky district, Ludic Karelians from the Pryazhinsky district of the Republic of Karelia, and Tver Karelians from the Tver Oblast), Veps from the Prionezhsky district of Republic of Karelia, Ingrian Finns from the Hatsina district, Ingrians from the Soikkola and Lower Luga areas in the Kingisepp district, and Western Votes from the Joenperä/Krakol’je area in the Kingisepp district of the Leningrad Oblast (N_Ychr_ = 357, N_AUTOS_ = 123, see Appendix A). Their biological samples were collected during the 2014–2020 fieldwork supervised by Prof. Elena Balanovska and Prof. Oleg Balanovsky and according to the methodology described in [42]. Written informed consent was obtained from all the donors. Biological samples were collected from unrelated individuals whose ancestors from at least three previous generations represented the studied population. This field strategy and the active participation of linguists and ethnographers in the fieldwork ensured the broadest possible coverage of the local populations in their diversity and allowed us to create representative (subtotal) sample sets even for such tiny minorities as Votes and Ingrians. The four Finnic groups were also compared to selected neighboring Russian populations (N = 763; see Appendix A).

### 2.2. Molecular Genetic Analysis

DNA was isolated from the samples of venous blood or saliva on a QIAsymphony SP instrument or, alternatively, by phenol-chloroform extraction preceded by digestion with proteinase K. For sample preparation, we used a Nanodrop 2000 spectrophotometer, a Qubit 4.0 fluorometer, and a QIAgility workstation.

The samples were genotyped for Y-chromosome markers using OpenArray technology, a QuantStudio 12 Flex Real-Time PCR system, TaqMan probes, and a custom panel of the most informative SNP markers. We selected 80 SNPs that best characterize the Y haplogroup spectrum of Finnic populations, including 20 SNPs that were further used for the phylogeography analysis of haplogroup N3a4 (Appendix A). The ages of the haplogroups provided in this article were borrowed from YFull [43] if not specified otherwise.

Genome-wide data were generated using an Infinium OmniExome BeadChip Kit (Illumina) and an iScan microarray scanner. The primary analysis and quality control were done in GenomeStudio v2011. 1. All the samples had a call rate of at least 0.99. The genome-wide data obtained will be made available on http://xn--c1acc6aafa1c.xn--p1ai/?page_id=36641 (accessed on 27 November 2024).

Differences in the sample sizes for the Y-chromosome and genome-wide arrays (see Appendix A) are related to the number of markers. For the Y-chromosome analysis, a larger sample size is used (tens of samples per population), but only one ancestral line is studied. The genome-wide array we used provides information on 4 million SNPs (inherited from all ancestors) for each individual. Therefore, when comparing populations where each sample is characterized in such detail (that it is close in information content to a whole genome analysis), samples from several individuals are sufficient. For example, in one of the first studies based on genome-wide panels (~650,000 SNPs), each of the regions of Europe was represented by 3 samples [44].

### 2.3. Statistical and Bioinformatic Analyses

The pairwise matrix of Nei’s genetic distances (Appendix A) was computed in the original DJ software (version 0.11 beta) [45] from the frequencies of 25 Y-chromosome haplogroups that are polymorphic in the studied populations (Appendix A). The multidimensional scaling plot of genetic distances was created in Statistica 7.0 (StatSoft©).

Genotypes of the Finnic populations and their Russian neighbors (N = 123; Appendix A) were subjected to principal component analysis (PCA). Before the calculation, the data were filtered: polymorphisms read in less than 95% of the samples (geno 0.05), and those occurring with a minor allele frequency of less than 1% (maf 0.01) were excluded. Samples in which less than 90% of the polymorphisms were read were excluded (mind 0.1). The exclusion of closely linked polymorphisms (r2 > 0.2) was carried out using the moving window method, including 1500 polymorphic sites at each step and shifting by 150 polymorphic sites per step (indep-pairwise 1500 150 0.2). The calculation of the values of the principal components was carried out using the smartPCA module from the EIGENSOFT software package version 7.2.1; the results were visualized using the matplotlib and seaborn libraries for the Python language [46].

Ancestral components were inferred for the same populations and samples that were included in PCA. ADMIXTURE modeling [47] at K = 2 to K = 23 was carried out for the same populations and samples as the PCA. We filtered the dataset using PLINK v1.9 with the following parameters: (a) a genotype missingness threshold of 0.05 (geno 0.05) to exclude poorly genotyped markers, (b) a minor allele frequency threshold of 0.05 (maf 0.05) to exclude low-frequency markers, and (c) a sample missingness threshold of 0.1 (mind 0.1) to remove poorly genotyped samples. Linkage disequilibrium pruning was performed with a window size of 50 SNPs, a step size of 10 SNPs, and an r^2^ threshold of 0.1 to exclude linked markers (LD 50, 10, 0.1). Samples with relatedness up to the third degree (kinship coefficient ≥ 0.0884) were excluded using the KING software (version 2.2.4) . After applying these filters, 100,211 markers remained in the dataset. The ADMIXTURE analysis was performed with 10-fold cross-validation.

### 2.4. Cartographic Analysis

Genetic geographic maps were created using the original GeneGeo software (version 2.79) developed under the supervision of Elena Balanovska and Oleg Balanovsky. The maps were constructed from the genotyping data and information from the offline Y-base database previously developed under the supervision of Oleg Balanovsky. Frequency distribution maps for the studied Y-chromosome haplogroups were created using the average weighted interpolation procedure with a search radius of 1200 km and a weight function inversely proportional to the cube of the distance [48]. The maps of genetic distances were constructed for 29 Y-chromosome haplogroups of the studied populations. The cartographic method allowed us to include information about the frequencies of these haplogroups in the populations outside Russia in the comparison dataset. First, the interpolation maps were built for each haplogroup, and then genetic distances from each studied Finnic population to all comparison groups were calculated and projected onto the map.

## 3. Results

### 3.1. Genetic Diversity of Russia’s Finnic Groups by Y-Chromosome

There are 18 polymorphic Y-chromosome haplogroups in the gene pool of the analyzed Finnic populations of Russia (Appendix A). The most common of them are I1, N3a3, N3a4, and R1a (Appendix A). The contribution of the haplogroup I1 peaks in Northern Karelians, Veps, and Ingrians (21–25%) and is high in Ingrian Finns (12%). The haplogroup N3a3 accounts for one-fifth of the overall Y-chromosome diversity of Tver Karelians (19%) and occurs at high frequencies in Ingrians, Votes, Veps, and Northern Karelians (9–13%). The haplogroup N3a4 constitutes one-third of the Y-chromosomal gene pool of Russia’s Finnic-speaking populations, prevailing in Ingrian Finns (67%) and Northern Karelians (50%). Its contribution to the gene pools of Ludic, Livvi, Tver Karelians, and Ingrians ranges from 21% to 38%. The haplogroup R1a, including its branch R1a-M458, is the dominant haplogroup in Votes (73%) and in Karelian Ludic (60%) and represents one-third of Y-chromosome diversity in Livvi and Tver Karelians, Veps, and Ingrians (Appendix A).

The studied populations differ noticeably in the diversity of the four main haplogroups and their frequencies (Appendix A) both between each other and from the averaged sample (combining all the studied Finnic groups, N = 357, Appendix A). The gene pools of the Finnic populations were compared to each other and to some neighboring Russian populations across the entire haplogroup spectrum using a matrix of pairwise Nei’s genetic distances (Appendix A). The data was visualized on a multidimensional scaling plot shown in Figure 2.

In the constructed genetic space (Figure 2), the populations are grouped in four clusters, except for two Northern Russian groups: the Pomors of the Summer and Winter Coasts previously described in [49].

The “Northern” cluster is formed by Northern Karelians and Ingrian Finns. Closer to the center is the “Ingrian-Veps” cluster that comprises Ingrians, Veps, and Northern Russians (the Pomors of the Onega coast and the population of the Krasnoborsk district in the Arkhangelsk region). The “Karelian” cluster consists of the Karelian populations (Ludic, Livvi, Tver Karelians) and of the Yaroslavl Russians of Mologa described in [50]. 

The “Southern” cluster at the bottom is the most heterogeneous: it comprises the populations of Votes and of the Russians of Novgorod, Yaroslavl, Pskov, and Tver regions. Inside the “Southern” cluster (Figure 2), two subclusters (α and β) are well distinguishable by their genetic distances (Appendix A). Subcluster α includes Votes and the two Russian populations geographically closest to the studied (westernmost) Votes: the Pskov Russians of Porkhov and the Russians of the Western Tver region (Selizharovsky and Torzhok districts). The more heterogenous subcluster β consists of three Novgorod Russian populations, the Yaroslavl (other) population, and the eastern population of Tver Russians (Kashinsky district). The Yaroslavl (other) population unites samples from different areas of the Yaroslavl region, with the exception of the descendants from Mologa. The population of the Pskov Russians of Ostrov lies at the cluster’s border and is close to only two Novgorod populations in our sample: Antsiferovo and Lubytino (further details are available in [51]).

The populations inside the clusters are close in terms of haplogroup frequencies. However, the haplogroup contributions differ between the clusters. The leading haplogroups of the “Northern” cluster are I1 and N3a4; their cumulative contribution is 77% (18% and 59%, respectively). The “Karelian” cluster is dominated by haplogroups N3a3, N3a4, and R1a, which together make up to 82% of the cluster’s Y-chromosomal gene pool (10%, 28%, and 44%, respectively). Haplogroups I1, N3a3, N3a4, and R1a prevail in the “Ingrian-Veps” cluster, constituting 81% of its gene pool (19%, 9%, 25%, and 28%, respectively). The most frequent haplogroups in the “Southern” cluster are N3a3 and R1a (11% and 56%, respectively; the cumulative contribution is 67%.

On the whole, the spatial arrangement of the populations on the plot is consistent with their geographical distribution, and the North–South vector is better defined than the West–East vector. However, there are three populations that do not comply with the principle of geographical clustering. First, the populations of Votes and Ingrians are geographical neighbors, and yet they are separated by a substantial distance on the plot (đ = 0.70). Second, Veps bear stronger genetic similarity to Northern Russians (the Pomors of the Onega and the Summer coasts and of Krasnoborsk; 0.08 < đ < 0.18) than to their closest Finnic-speaking geographical neighbors (Livvi and Ludic Karelians, 0.22 < đ < 0.27).

The third exception is two Russian groups, specifically the Northern Russians of Krasnoborsk, who live much further to the south from the Pomors of the Onega coast and from Veps, and the Pskov Russians of Ostrov and Porkhov, who live in close proximity, and yet their gene pools are significantly different from each other (for further details see [45,51]).

The boundaries between the four clusters are not strict. This becomes evident when the matrix of genetic distances (Appendix A) is compared to the multidimensional scaling plot (Figure 2). The most illustrative example is Ingrians: the gene pools with the most affinity to Ingrians are not limited to the “Ingrian-Veps” cluster (0.06 < đ < 0.17) but also fall into the “Northern” (0.07 < đ < 0.13) and “Karelian” clusters (Livvi, Ludic, Tver Karelians, Mologa Russians: 0.13 < đ < 0.15). Therefore, it would be more accurate to place Ingrians between these three clusters. This suggests high genetic variation within the Ingrian population and a possible genetic contribution of these three clusters to the Ingrian gene pool.

Similarly, Ludic Karelians are as genetically close to the geographically proximate Livvi population (đ = 0.11) as they are to the Central Russians from the “Southern” cluster (Novgorod, Kashin Tver, and the generic Yaroslavl populations, 0.09 < đ < 0.12).

### 3.2. Four Y-Chromosomal Genetic Patterns in the Context of Northern Europe

For each population of Russia’s Finnic groups, the closest related gene pools of Northern Europe were analyzed using the genetic cartographic method. The resulting maps of Nei’s genetic distances (Appendix A) were arranged into four groups, which will also be referred to as “Northern”, “Ingrian-Veps”, “Karelian”, and “Southern” patterns (Figure 3A–D). Within each pattern, the similarity between the maps is the greatest (the correlation coefficient is the highest; see Appendix A). These four patterns correspond to the clusters on the multidimensional scaling plot (Figure 2).

The “Northern” pattern (Figure 3A) is based on the averaged maps of Nei’s genetic distances for Ingrian Finns and Northern Karelians (Appendix A). This pattern is also represented by the populations of Northern and Eastern Finland (0.03 < đ < 0.18) and by Ingrians (đ = 0.10).

The “Ingrian-Veps” pattern (Figure 3B) reaches far eastward and westward. In the west, it is found in the populations of Finland, except for its northern regions (0.06 < đ < 0.17), and, with less intensity, in some populations of Sweden (0.16 < đ < 0.18) and Norway (0.16 < đ < 0.17). In the east, the pattern is represented by the populations of the Russian Pomors of the Onega coast (đ = 0.07) and of Krasnoborsk Russians (đ = 0.12). It is less common in Vologda Russians (đ = 0.17), the Yaroslavl Russians of Mologa (đ = 0.18), and in Tver Karelians (đ = 0.18). The correlation coefficient between the distance maps for Ingrians and Northern Karelians is high (r = 0.91, Appendix A). However, the maps for Veps and Northern Karelians are different, so these three populations cannot be clustered under one pattern. Veps (Appendix A) are fairly distant from the populations of Eastern and Northern Finland and from Northern Karelians. In turn, Ingrians are distant from Novgorod, Pskov, Vologda, and Kostroma Russians and from Winter Coast Pomors (Appendix A). Despite these differences and the low similarity between the gene pools of Veps and Ingrians (đ = 0.17), the correlation coefficient between their genetic distance maps is higher than with other populations (r = 0.94) (Appendix A).

The “Karelian” pattern (Figure 3C) is based on the genetic distance maps for three southern populations: Livvi, Ludic, and Tver Karelians (Appendix A). The “Karelian” pattern is widespread: it is observed in the populations of Eastern Finland (0.08 < đ < 0.19) and in Ingrians (đ = 0.18) in the west, in Northern Russians (Arkhangelsk Russians from the Leshukonsky district and Vologda Russians, đ = 0.12) in the east, and in Central Russians (Novgorod, Ivanovo, Vladimir, 0.16 < đ < 0.18) in the south. There is a certain similarity between genetic distance maps for Ludic Karelians and Votes (Appendix A, r = 0.91). However, the mutual genetic similarity between the three Southern Karelian populations is greater than their similarity to other Finnic or Russian populations (0.94 < r < 0.98). Differences between the genetic distance maps for Votes, Tver Karelians, and Livvi Karelians prevent these populations from being clustered under one pattern.

The “Southern” pattern is represented by Votes only, their map being very different from the rest of the studied populations (Figure 3D). This pattern is shifted more to the south in comparison with the other Finnic groups under study. The only two Northern Russian groups showing some similarity to Votes are the Arkhangelsk Russians from Leshukonsky district (đ = 0.14) and Vologda Russians (đ = 0.16). Votes bear genetic similarity to Central and Southern Russians (Novgorod, Vladimir, Ivanovo, Smolensk, Bryansk, Tula, Ryazan, and Lipetsk regions, 0.04 < đ < 0.10), Belarusians (0.04 < đ < 0.05), and the populations of the Volga-Ural region (Erzya Mordvins, Maris, Bashkirs, Volga Tatars, 0.04 < đ < 0.20).

The averaged map of genetic distances (Figure 3E) covers the habitats of Russia’s Finnic groups and the populations of Eastern Finland. The map in Figure 3F shows the area where the four patterns overlap (the red circle), which may be the eastern part of the homeland of the Proto-Finnic gene pool.

### 3.3. Phylogeography of the Haplogroup N3a4 in Northeastern Europe

We found that N3a4-Z1936 is the most common haplogroup in the gene pool of the studied Finnic populations: its frequency varies from 20% in Votes to 67% in Ingrian Finns and averages 33% (Appendix A). We have analyzed its intragroup variation and geographical distribution to get an insight into the population dynamics within its geographical boundaries and estimate the timelines of demographic changes. Therefore, we screened the N3a4 carriers representing 63 populations from Northeastern Europe and the Volga-Ural regions for 20 SNPs specific to N3a4 branches (N_N3a4_ = 573 haplogroup carriers from the general dataset, N = 8641).

The diversity of N3a4-Z1936 in Ingrian Finns is represented by its three branches: the major subgroup N3a4-Z1927 and the minor subgroups N3a4-YP5259 and N3a4-SK1485 (Appendix A). The sub-haplogroup N3a4-YP5259 (~3.5 ky) and its branch N3a4-Z35275 (~2.2 ky) occur in Veps (6%), Livvi and Ludic Karelians (2% and 2%, respectively). The sub-haplogroup N3a4-SK1485 (~3.4 ky) and its branch N3a4-YP6094 (~2.8 ky) occur only in Veps (4%). Overall, N3a4-YP5259 and N3a4-SK1485 account for two-thirds of N3a4 frequencies in Veps (10% out of 15%; Appendix A).

The main branch N3a4-Z1927 (~2.4 ky, varies from 4% in Veps to 67% in Ingrian Finns) is represented by subgroups N3a4-Z19825 and N3a4-CTS4329 in the Finnic groups in question. To analyze the phylogeography of the entire N3a4-Z1927 branch, we constructed 15 genetic geographic maps (Figure 4).

The sub-haplogroup N3a4-Z19825 (~1.9 ky) is widespread in Northeastern Europe: apart from Finnic populations, it occurs in Northern (Arkhangelsk and Vologda) and Central (Novgorod and Pskov) Russian populations (Figure 4). Below, we provide a description of its three branches: N3a4-Z19833, N3a4-Z4747, and Z19825*.

N3a4-Z19833 is the youngest branch (~1.3 ky) with the most limited geographical distribution: it occurs only in Northern and Tver Karelians and in Ingrian Finns (3–5%).

The branch N3a4-Z4747 (~1.9 ky) is close to its parental haplogroup N3a4-Z19825 in terms of age. Three N3a4-Z4747 sub-branches (Z4747*, Z1941*, and Z1937) date back to roughly the same period (~1.7 ky and ~1.9 ky, see Figure 4). Their geographical distribution resembles puzzle pieces that fit together to form the area of N3a4-Z4747, overlapping only in Ingrian Finns and in the Pomors of the Onega coast. N3a4-Z4747* occurs at low frequencies (2–3%) in Ingrian Finns, Livvi Karelians, and the Pomors of the Onega coast. The branch N3a4-Z1941* has a more extensive geography and higher frequencies, occurring in Ingrians (18%), Ingrian Finns (8%), Pskov Russians of Ostrov (3%), Livvi (2%) and Ludic Karelians (2%), and in Northern Russian Pomors (1–5%). N3a4-Z1937 is spread across vast territories: its frequencies peak in Ingrian Finns (24%) and decline in Northern Karelians (8%), the Russian Pomors of the Summer Coast (4%), and a Novgorod population of Kabozha (3%).

The branch N3a4-Z19825* is a root cluster (full nomenclature: Z19825(xZ19833,Z4747)) and presumably comprises a group of yet unknown subbranches. This assumption is based on the vast geography of N3a4-Z19825*, which occurs not only in Karelians and Veps but also in the Northern Russians from Arkhangelsk and Vologda regions. Further research of the branch Z19825* might lead to the discovery of its lineages shared by Northern Russians and the populations of Karelia, as well as to the region-specific variants that have emerged in the past 2 ky, i.e., throughout the existence of N3a4-Z19825.

The sub-haplogroup N3a4-CTS4329 (~2.1 ky) is more specific to Finnic populations, specifically for Northern Karelians (31%), Ingrian Finns (19%), Ingrians (16%), and Livvi Karelians (8%). Among the populations of Russia considered here, it has peak frequencies in the Yaroslavl Russians of Mologa (5%) and Northern Russians, as well as in the populations of the Volga region (Erzya Mordvins, Maris, Mishari Tatars; 1–3%).

Within N3a4-CTS4329, we analyzed three branches: the main branch, N3a4-Y125466, and two rare branches, N3a4-Y31248 and N3a4-CTS4329*. The geographic diversity of N3a4-Y125466 (~2 ky) is largely represented by its sublineage N3a4-CTS3223 (~1.7 ky), which has the highest frequencies (Appendix A) in Northern Karelians (31%), followed by Ingrian Finns (8%), Livvi Karelians (4%), and Ingrians (4%). Other variants (cluster N3a4-Y125466(xCTS3223) are occasionally found in Ingrian Finns (5%), the Russian Pomors of the Winter Coast (3%), and in Erzya Mordvins (2%). The rare branch N3a4-Y31248 (~1.6 ky) is limited to Livvi Karelians, Ingrian Finns, and the Russians of Mologa (3–5%). The cluster N3a4-CTS4329* (full nomenclature: N3a4-CTS4329(xY125466, Y31248)) is found only in Ingrians (11%) and in the Northern Russians of Krasnoborsk (3%).

Therefore, the diversity of N3a4 in the studied Finnic-speaking populations is largely represented by nine parallel branches that date back ~1.3 to ~2.1 kya. Four of them have a vast geography (Z19825*, Z1941*, Z1937, and CTS3223), and the rest occur as isolated islands.

### 3.4. Autosomal Gene Pool Structure for Russia’s Finnic Populations

PCA was carried out for the populations of Northern Karelians, Ludic, Livvi, Tver Karelians, Veps, Ingrian Finns, Votes, and Ingrians, as well as for the Central Russians from the Pskov region (Ostrov and Porkhov), the Novgorod region (Lubytino, Kabozha, general population), and the Yaroslavl region (Mologa and the generic Yaroslavl population), see Appendix A. Because the PCA plots were based on the data generated from individual genomes, we were able to analyze how the centroids (the centers of the ‘population clouds’) are positioned relative to each other and to assess the degree of intrapopulation variation.

The structure of the genetic space is reflected on the PCA plots for the principal components (Figure 5). PC1 (Figure 5A; *x*-axis) corresponds to the distribution of the studied populations from south to north (and from the Slavic-speaking to the Finnic-speaking groups). The groups are spread horizontally in the south–north scale from the Russians of Pskov, Novgorod, and Yaroslavl regions towards Karelians, Veps, Votes, Ingrians, and Ingrian Finns. PC2 (Figure 5A; *y*-axis) describes their variation in the vertical dimension from the east (Karelia’s populations, the Russians of Novgorod and Yaroslavl regions) to the west (Ingrian Finns, Ingrians, Votes, Pskov Russians). PC3 (Figure 5B) is comparable to PC2 in terms of its informative value but partially contradicts the west-to-east distribution from plot A.

Finnic groups and the adjacent populations of Russians form four clusters: “Karelia” (all Karelian populations and Veps), “Ingria” (Ingrians and Votes), “Pskov” (Russian populations of Ostrov and Porkhov districts in the Pskov region), and “Novgorod-Yaroslavl” (Russian populations of the Novgorod and Yaroslavl regions). Ingrian Finns are located between the “Karelia” and “Ingria” clusters on the PC2 axis (Figure 5A) and clustered with Ingrians and Votes (“Ingria”) on the PC3 axis (Figure 5B).

In the PC1*PC2 space (Figure 5A), the greatest variation is observed within the “Karelia” cluster. Of all populations constituting this cluster, Veps, which are represented by a small number of samples (N = 9), demonstrate a degree of variation comparable to that of the general Karelian population (from Northern to Tver Karelians, N = 25, Appendix A). The gene pools of Ludic and Livvi Karelians are in the center of the cluster, inside a triangle formed by Northern Karelians, Veps, and Tver Karelians. On the PC1*PC3 plot (Figure 5B), the structure of the “Karelia” cluster is retained, and the apices of the triangle (Northern Karelians–Veps–Tver Karelians) are better defined. On the PC2*PC3 plot (Figure 5C), the size and shape of the “Karelia” cluster are the same as on the PC1*PC2 plot (Figure 5A). However, the populations are distributed differently within the cluster: Veps are at one extreme, whereas Northern and Tver Karelians are at the opposite extreme (coming closer to each other on the level of PC3). Ludic and Livvi Karelians are located between the two extremes.

The degree of variation and the size of the “Ingria” cluster in the PC1*PC2 space (Figure 5A) is determined by the intrapopulation variation of Ingrians. Votes are projected to the center, shifting towards Pskov Russians at their periphery. On the PC1*PC3 plot (Figure 5B), the “Ingria” cluster includes Ingrian Finns, which makes it comparable to the “Karelia” cluster in terms of size. Ingrian Finns are clustered densely, only shifting slightly towards the “Karelia” cluster. On the PC2*PC3 plot (Figure 5C), “Ingria” and the “Novrgorod-Yaroslavl” clusters come closer together, with the Ingrian Finnish gene pool gravitating to the latter.

The “Pskov” cluster stands out in the degree of its intracluster variation, which is significantly greater than in the “Novgorod-Yaroslavl” cluster and is comparable to that in the “Karelia” and “Ingria” clusters. The clouds of “Pskov” populations overlap, but only Porkhov genomes gravitate to the other three clusters (Figure 5A). On all PCA plots, the “Pskov” cluster is distinctly separated from its Russian neighbors (Figure 5) by distance, which is the same or even greater than that of the two Finnic clusters. The “Novgorod-Yaroslavl” cluster is the most compact of all. Within the “Novgorod-Yaroslavl” cluster, the centroids of three Yaroslavl and two Novgorod populations and the majority of individual genomes are placed very close to each other.

PCA reflects the main trends in the gene pools of the Northeastern European regions studied. For a more detailed analysis, we resorted to the ADMIXTURE-based ancestral component modeling.

### 3.5. Ancestral Component Modeling

ADMIXTURE estimations were carried out at K = 2 to K = 23 (the number of ancestral components) for the same populations and samples that were included in PCA. The two most informative models at K = 6 (basic) and K = 22 (detailed) were selected to analyze the gene pool of the Finnic groups and the adjacent Russian populations (Figure 6, Appendix A).

In the basic model at K = 6, over 80% of the gene pool of each population is represented by a sum of two major ancestral components, k4, and k3, shown in red and green, respectively, in Figure 6A. The average contributions of these components differ remarkably between the Finnic(k4 = 64%, k3 = 26%, 2.5:1, respectively) and the Central Russian populations (k4 = 40%, k3 = 50%, 1:1.3, respectively). Among the Finnic groups, the proportion of the ancestral component k4 is considerably higher in the populations of Ingria, a historical Swedish province in Northwestern Russia (85% in Ingrian Finns, 81% in Ingrians, 67% in Votes), than in the populations of Karelia (63% in Ludic, 60% in Veps, 57% in Northern Karelians, 55% in Livvi, 44% in Tver Karelians; see Appendix A). Among the Russian populations, the contribution of k4 is the greatest in Novgorod Russians (50% in Lubytino and 46% in Kabozha, 44% in the general population) and the lowest in Pskov Russians (29% in Porkhov and 34% in Ostrov populations). The ancestral component k3 prevails in Pskov Russians (60–61%) and demonstrates a minimal contribution in Novgorod Russians (41–45%). Among the Finnicpopulations, its contribution is the greatest in Tver Karelians (40%), varies from 25% to 32% in Northern Karelians, Livvi, Ludic, Veps, and Votes, and is the smallest in Ingrians and Ingrian Finns (17% and 11%, respectively, Appendix A).

The proportion between k4 and k3 (Figure 6) is similar to the PC1 trend (Figure 5A) in the studied groups. The Finnic populations are characterized by a significant contribution of k4 and are located in the region of positive PC1 values, whereas the Russian populations carry a higher proportion of k3 and are located in the region of negative PC1 values.

The rest of the ancestral components represent a small fraction of the gene pool of the studied populations (3–16%) in the basic model at K = 6 (Figure 6A). Components k1 and k5 have a significant presence in the populations of Karelia and are almost non-existent in Ingrians, Votes, and Ingrian Finns, whereas the contribution of k2 is the greatest in Russians, especially from the Novgorod and Yaroslavl regions. On the whole, the components k4, k1, and k5 can be regarded as characteristic of the Finnic populations, and the components k3 and k2 are characteristic of the Central Russians studied in this work.

The detailed model built at K = 22 (Figure 6B, Appendix A) contains seven informative ancestral components that cumulatively constitute 83–97% of the studied gene pools. Out of these, four are major (k4, k3, k22, k10) and three are minor (k16, k18, k15).

The major components k4 and k3 reproduce the trend revealed in the basic model at K = 6, but with sharper differences between the Finnic-speaking (k4 = 49%, k3 = 6%; 8:1) and the Russian populations (k4 = 10%, k3 = 38%; 1:4). These differences result from the presence of two new components in the detailed model: k22 and k10. The ancestral component k22 (shown in purple in Figure 6B) occurs in all the studied populations (at 16% on average), has the highest prevalence in the gene pools of Karelians and Veps (~26%), is 2.5 times less frequent in the Russian populations and four times less frequent in Ingrian Finns, Votes, and Ingrians (Appendix A). The component k10 (shown in blue in Figure 6B) is not present in Karelians and Veps but constitutes up to one-fifth (19%) of the Votic, Ingrian, Ingrian, Finnish, and Russian gene pools.

The distribution of the minor components k15 and k16 (shown in beige and brown, respectively, in Figure 6B) in the detailed model is similar to that of k5 and k1 in the basic model at K = 6. The distribution of k15 (shown in beige) is less uniform than that of k5 in the basic model. At K = 22, the contribution of k15 is higher in Veps, Ludic, and Livvi Karelians, and Novgorod and Yaroslavl Russians (Figure 6B) but declines considerably in Northern and Tver Karelians and has almost zero value in the populations of the historical Ingria. The contribution of k16 (shown in brown) in the detailed model declines, as compared to that of k1 in the basic model, in all populations except for Tver Karelians and the Russians of Mologa and Porkhov. The third minor component, k18 (present only in the detailed model), is more characteristic of the Karelian populations: 5% (peak) in Northern Karelians, 3% in Tver and Livvi Karelians, 2% in Veps, and 1% in Ludic. This is higher than in the Ingrian (0–1%) and Russian (0–2%) populations (Appendix A).

The ratio of the ancestral components in both models suggests the clustering of the populations into three large blocks (Figure 6): all Karelians and Veps (“Karelia”), Ingrian Finns, Votes, and Ingrians (“Ingria”), and the Russians from the Novgorod, Yaroslavl and Pskov regions (“Central Russians”). In the basic model, the contributions of k4 vs. k3 are 56% vs. 30%, respectively, in the group of the “Karelia” populations, 77% vs. 19%, respectively, in the group of the “Ingria” populations, and 40% vs. 50%, respectively, in the group of “Central Russians” (Appendix A). In the detailed model (Appendix A), the red k4 (45%) and the purple k22 (26%) have the highest frequencies in the “Karelia” block. In the group of the “Ingria” populations, the most frequent are the red k4 (57%), the blue k10 (18%), and the green k3 (12%) components. In the group of “Central Russians”, who are geographically close to the studied Finnic groups, the most prevalent are the green k3 (38%), the blue k10 (20%), the purple k22 (12%), and the red k4 (10%) components.

In both models, “Ingria” has the highest proportion of k4, which makes it similar to “Karelia”. In the detailed model, “Ingria” is characterized by a substantial presence of k10, which is similar to the populations of Russians. “Karelia” is intermediate between the other two blocks in the basic model and has the highest k22 frequencies and the greatest sum of the three minor components (k16, k18, k15) in the detailed model. Unlike the two Finnic-speaking blocks, the frequency of k3 reaches its maximum in the group of “Central Russians” in both models. However, at K = 22, “Central Russians” are similar to “Ingria” in the proportion of two major components, k10 and, to some extent, k22, while they are close to “Karelia” by the contribution of two minor components, k15 and k16.

## 4. Discussion

### 4.1. The Genetic Diversity of Finnic Populations

The autosomal composition of the populations of Karelia (Veps and Karelians) is more homogenous than their Y-chromosomal gene pool. The Y-chromosomal gene pool of Karelia is largely shaped by the contribution of three genetic components and is characterized by their interpopulation differences in the diversity of the N3a4 branches. Differences between Northern and Southern (Ludic and Livvi) Karelians can be explained both by their geographical isolation from each other (~400 km) and the history of their formation. They might also indicate a genetic kinship between Ludic Karelians and a vanished ancient population that has left its trace in the gene pool of Central Russians.

On the Y-chromosome level, the big genetic distance between Veps and the three Karelian populations but the affinity of Veps to Northern Russians may indicate that Veps and the Finnic-speaking populations of the Russian North, assimilated in the 16–19th centuries, have a common genetic background. Indeed, the disappeared Zavoloch Chud of the Pomorje area has often been linked to Veps [17] (pp. 115–116, 140).

Tver Karelians have lived ~500 km away from other Karelian populations during the past three centuries. An earlier study has demonstrated a similarity between the gene pools of Tver Karelians and the Karelian and Veps populations of Karelia [41]. Using an expanded panel of Y-chromosome haplogroups and a genome-wide SNP panel, our study demonstrates the genetic affinity of Tver Karelians to Livvi and, to some extent, to Ludic Karelians and Ingrians. This disagrees with historical records that suggest a common origin of Northern and Tver Karelians [52,53].

The Y-chromosome analysis reveals that Ingrian Finns are the closest to Northern Karelians and to Eastern Finns, including the Finns from the provinces of North Savo and South Savo, which are historically considered as a source of migration of “Savakot”, some of the ancestors of Ingrian Finns (Appendix A), to Ingria. The greatest diversity of lineages within N3a4 is observed among Ingrian Finns; the presence of these lineages in Ingrians and Karelians suggests a common descent. The analysis of autosomal data shows a genetic proximity between Ingrian Finns, Ingrians, and Votes, which may have resulted either from marriages between Ingrian Finns and Ingrian and Votic women or from the fact that a substantial proportion of the modern gene pool of Ingria descends from an original local population (possibly from Votes or local Slavic tribes). Given that Votes and Ingrians were Orthodox like Russians, the intermarriages between these three groups used to be historically common, while their marriages with Lutheran Ingrian Finns were rarer before WWII but still attested [54].

Despite being geographical neighbors, Votes and Ingrians occupy the opposite extremes of the genetic space constructed from Y-chromosomal data. The Votic gene pool is strikingly different from all the other Finnic groups considered due to a significant contribution of the haplogroup R1a. Votes fall into a separate distant cluster, together with the Central Russian populations. This may be due to gene drift, the small size of the group, or the initial dissimilarity between the Votic and Ingrian gene pools. Considering the lack of migratory activity among Votes, the third assumption might be plausible. 

The intrapopulation diversity of Ingrians is amazing, both in terms of autosomal (Figure 5) and Y-chromosomal markers. Being formally part of one cluster, Ingrians also gravitate towards another two (Figure 2), and their genetic distance map falls into the patterns: Ingrian, Northern, and Karelian (Appendix A). This is understandable because our Ingrian sample actually consists of Soikkola and Lower Luga Ingrians, which have quite different ethnic histories (to be explored in a separate study currently under preparation).

The structure of the autosomal and Y-chromosomal gene pools of the Finnic populations of Russia differs significantly. The autosomal gene pool of the Finnic groups can be broken down into two large blocks (Figure 5 and Figure 6) based on their geographic proximity: the populations of historical Ingria (Votes, Ingrians, and Ingrian Finns) and of Karelia (Veps and all Karelian populations, including the genetically close Tver Karelians). Such structure is detectable at the basic level of ancestral modeling (K = 6), which indirectly suggests its ancient origin, and can still be identified even in the detailed model (K = 22). In the autosomal genetic space, Central Russians are separated from the Finnic groups, although they fall into the same cluster with Votes in the Y-chromosome space. The Y-chromosomal gene pool of the Finnic populations is more diverse than the autosomal one. It spans most of the multidimensional genetic space, where the studied populations are distributed into four clusters (Figure 2), reproduced on the Northern European scale in the cartographic analysis (Figure 3A–D).

Therefore, our results support the formulated hypotheses only partially.

We do not observe a clear division of the Finnic populations studied into two groups, Votes and others (Karelians, Ingrian Finns, and Ingrians), according to any system of genetic markers. By the Y-chromosome, Votes are indeed distant from the other Finnic groups, but the differences among the latter are also great.

The similarity of Votes with Novgorod Russians is revealed only by the Y-chromosome and only with the modern Pskov Russians of Porkhov (formerly part of the Novgorod Republic and the closest geographic neighbors of Votes out of all the Novgorod Russian populations considered here). The same degree of similarity is observed between Votes and the Russians of the Western Tver region (Selizharovsky and Torzhok districts, see Appendix A).

A common genetic component associated with the historical group of Zavoloch Chud has been revealed by the Y-chromosome for Veps and the two groups of Northern Russians geographically distant from each other and from Veps (the Russian Pomors of the Onega coast and of Krasnoborsk Russians, see Figure 2). The Northern Russians autosomal gene pools remains to be explored.

The genetic relationships between Veps, Livvi, Ludic, Tver, and Northern Karelians also differ from what was expected. The autosomal gene pool of Veps is indeed closest to Livvi and Ludic Karelians, as expected. However, according to the Y-chromosome, Veps are quite distant from Livvi, Ludic, and Tver Karelians but even more distant from Northern Karelians (see Appendix A).

### 4.2. The Origins of the Haplogroup N3a4 for Finnic and Neighboring Populations

The structure of the autosomal gene pool depends on a significantly higher number of a person’s ancestors than the structure of the Y-chromosomal gene pool. Although the Y-chromosomal gene pool is more susceptible to the impact of a gene drift, it better retains the ancient ancestral components due to the patrilocality and the patrilinear inheritance of haplogroups through thousands of years.

The haplogroup N3a4-Z1936 is the primary haplogroup for the populations of three out of four clusters in the Y-chromosomal genetic space (Figure 2, Appendix A). In the gene pool of the Finnic groups, it is represented mostly by the sub-haplogroup N3a4-Z1927 (its ancestral branch B535/Z1934 was detected at high frequencies in Karelians, Veps, Finns, Saami, and Northern Russians in [40]). This means that the majority of N3a4-Z1936 carriers among the Finnic populations, as well as among some Northern and Central Russian populations, descend from a common ancestor who lived ~2.4 kya (for N3a4-Z1927 dating see [43]).

The haplogroup N3a4-Z1927 is represented by N3a4-Z19825 and N3a4-CTS4329 that date back ~2 kya. Both branches occur in at least one Karelian population and at least one of the populations from Ingria. These branches are also distributed throughout Finland with high frequency, 21.1%, and 16.6%, respectively [30]. The phylogenetic tree suggests a population growth in the communities of the N3a4-Z1927 carriers because there are new lineages within each of its branches. The lineages CTS3223 and Z1941 (each of which dates back ~1.7 kya) are found both in Karelia and among the populations of historical Ingria, as well as in Finland. Therefore, while the ancestor of N3a4-Z1927 lived about 2.4 kya, it seems that rapid population growth among his Finnic descendants occurred only about 1.7–2.0 kya. This growth chronologically corresponds to the archaeological period of the so-called “typical” Tarand graves (1st–4th c. AD), the “golden” age of the Proto-Finnic civilization [15]. However, the Tarand graves are the most typical for Estonia, while the Y-haplogroup N3a4 is relatively rare among Estonians [39]. Therefore, the alleged population growth among the carriers of N3a4-Z1927, about 1.7–2.0 kya during the Tarand time does not bear clear relation to the spread of the known Tarand culture. However, the area of distribution of the Finnic populations under consideration is not entirely migratory and includes the easternmost part of the Tarand culture (cf. also the area of intersection of the four Y chromosome clusters, highlighted in red in Figure 3F and possibly reflecting the eastern part of the homeland of the Proto-Finnic gene pool). Study of ancient DNA from Tarand graves in this easternmost Proto-Finnic corner could shed light on a possible connection between the Tarand culture and the spread of the Y-chromosome N3a4-Z1927 during that period.The results of this work allow us to outline at least three directions for further research. First, there is a need for a thorough analysis of genetic relationships between the Finnic populations and the Northern Russians, as the Y-chromosomal gene pools of the Pomors of Onega coast and the Russians of Krasnoborsk share quite a few genetic components with Ingrians and Veps (Pomors also bear similarity to Livvi Karelians). Second, the dramatic differences between Votic and Ingrian Y-chromosomal gene pools and the high intrapopulation variation of Ingrians necessitate further analysis aided by interdisciplinary data. Finally, the Siberian component, not addressed here, should be studied for the genetic pools of the Eastern Finnic populations in comparison with a larger panel of samples, including the populations of Siberia.

## 5. Conclusions

This study is the first comprehensive analysis of the gene pool of the Finnic populations of Russia, with the use of the two most informative systems of genetic markers. The Y-chromosomal gene pool of the studied Finnic groups is more diverse than their autosomal gene pool and is constituted by four genetic components. The “Northern” component prevails in Northern Karelians and Ingrian Finns, the “Ingrian-Veps” in Ingrians and Veps, the “Karelian” in Livvi, Ludic and Tver Karelians, and the “Southern” in Votes. The autosomal gene pool of these Finnic populations can be divided into two large categories based on the results of PCA and ADMIXTURE modeling: “Karelia” (Veps, Northern Karelians, Ludic, Livvi, and Tver Karelians) and “Ingria” (Ingrians, Votes, Ingrian Finns). In the detailed ADMIXTURE model (K = 22), the ancestral components k4 and k22 occur in all populations but prevail in the Finnic ones. The component k10 equally contributes to “Ingria” and to the group of “Central Russians”, and k3 peaks in “Central Russians”. 

The phylogeographic analysis of the haplogroup N3a4-Z1927 suggests that the Finnic groups studied and some Northern and Central Russian populations had a common ancestor ~2.4 kya, but a population growth occurred among the carriers of N3a4-Z1927 only much later, ~1.7–2 kya, in the period of the “typical” Tarand graves when the Finnic civilization in general flourished. However, the fact that Estonia was one of the centers of the Tarand culture, but N3a4 is rare among modern Estonians indicates that more research is needed to study a possible relation of this alleged population growth and the spread of the Tarand culture. 

Our results, which showed high internal heterogeneity of Ingrians, a relatively isolated position of Veps among the Finnic groups studied, and an affinity between Veps and Russian Pomors, necessitate further interdisciplinary study of all these groups and a comparison of Eastern Finnic populations to a bigger set of samples which would include also Siberian populations.

## Figures and Tables

**Figure 2 genes-15-01610-f002:**
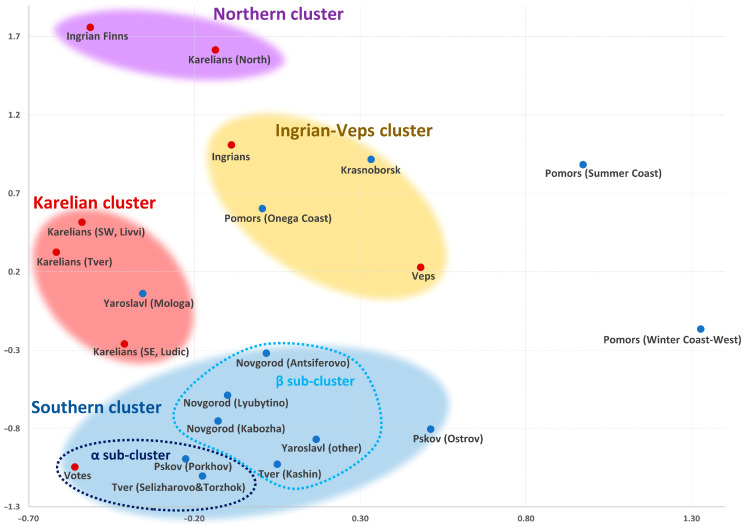
Position of Finnic populations in the genetic space of Northeastern Europe (multidimensional scaling plot for 18 populations based on 25 Y-chromosomal haplogroups, stress 0.07, alienation 0.09).

**Figure 3 genes-15-01610-f003:**
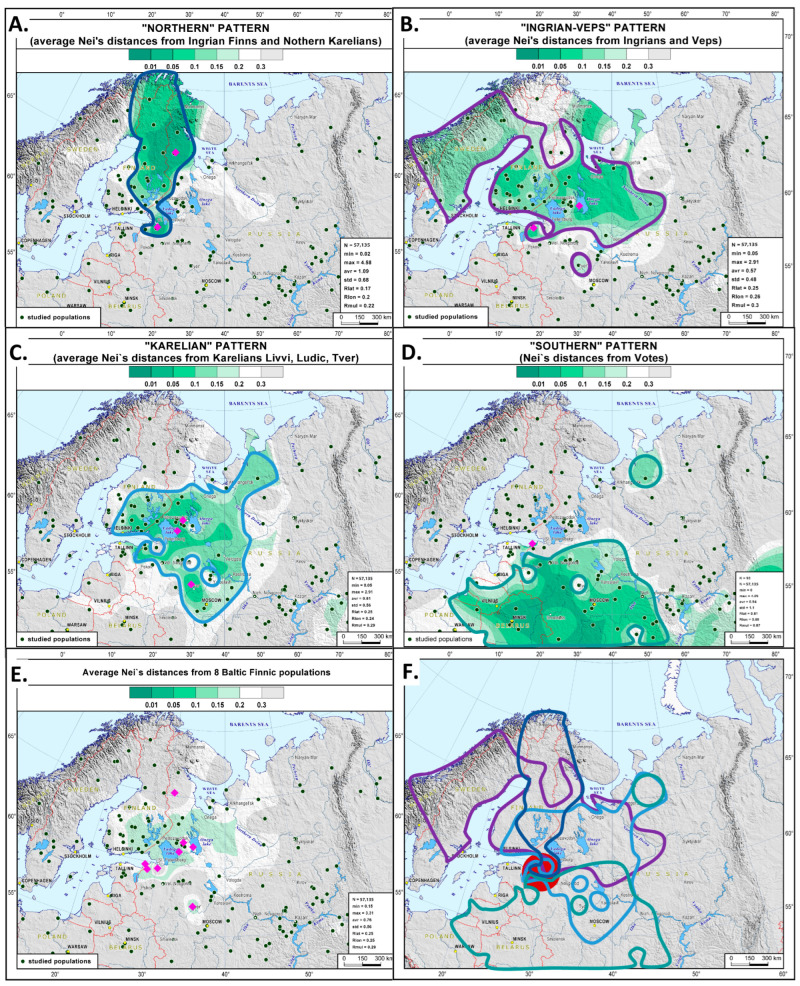
Geographical distribution of the four main patterns of the studied Finnic gene pools in Northern Europe (**A**–**D**). The map (**E**) is an averaged map showing the four patterns, while the map (**F**) shows the overlap of four patterns.

**Figure 4 genes-15-01610-f004:**
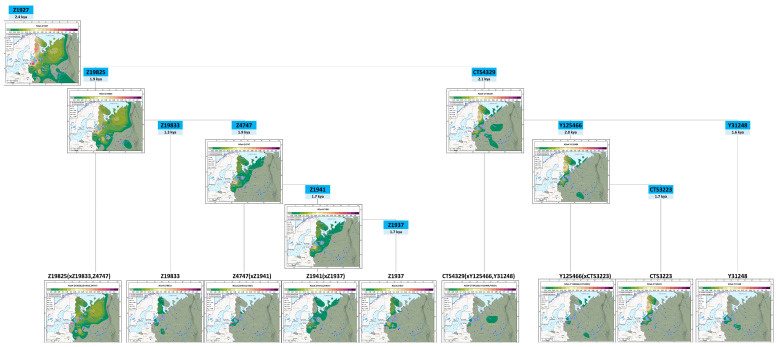
Phylogeny, dating, and distribution maps for the haplogroup N3a4. The maps are arranged according to the phylogenetic tree and the TMRCA-agess of N3a4 branches (see [43]).

**Figure 5 genes-15-01610-f005:**
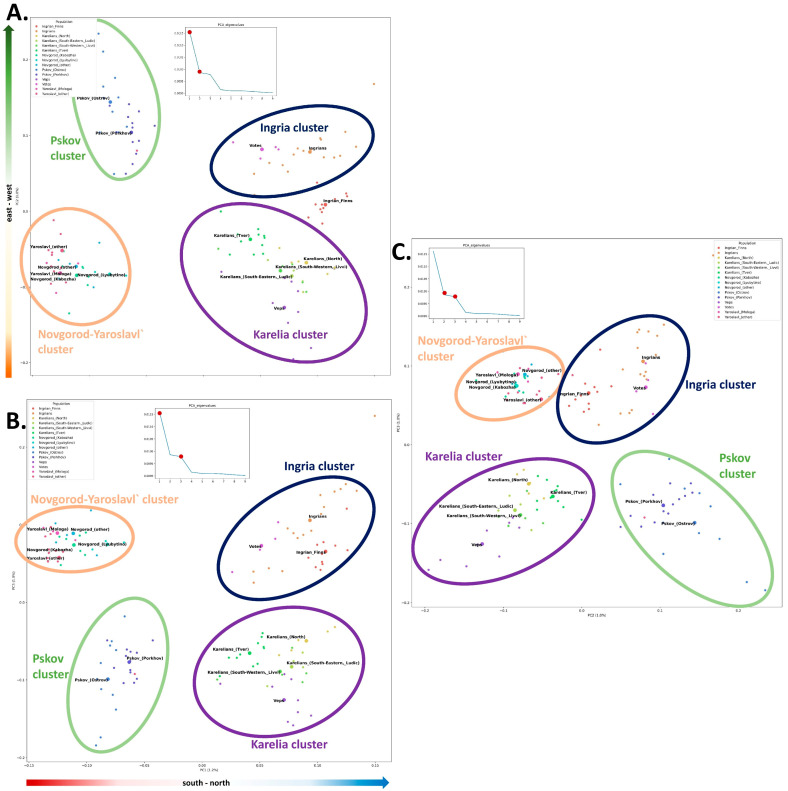
Gene pools of Finnic populations studied and of some adjacent Russian groups in the principal component (PC) genetic space constructed from autosomal markers: (**A**) PC1*PC2, (**B**) PC1*PC3, (**C**) PC2*PC3. Small dots represent individual genomes; big dots represent the centroids (centers of population clouds).

**Figure 6 genes-15-01610-f006:**
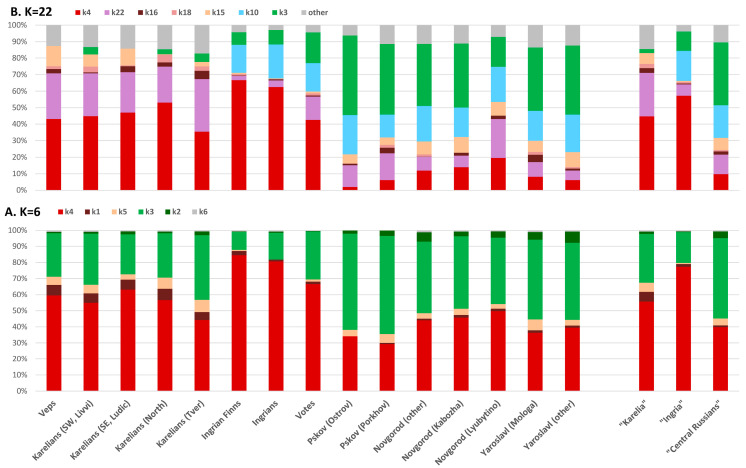
Average values of ancestral components in the studied populations at K = 6 (**A**) and K = 22 (**B**).

## Data Availability

Data and materials are available in Appendix A and on http://xn--c1acc6aafa1c.xn--p1ai/?page_id=36641 (accessed on 27 November 2024).

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
