# Peer review of "The Finnic Peoples of Russia: Genetic Structure Inferred from Genome-Wide and Y-Chromosome Data"

_genes, 2024, doi:10.3390/genes15121610_

Round 1

Reviewer 1 Report

Comments and Suggestions for Authors

In this manuscript, Agdzhoyan and colleagues reported their results of population genetics for the Baltic Finnic populations using more diverse samples performing both the Y-chromosomal and autosomal gene pool analysis, which led to better resolution for the relationships for the Baltic Finnic populations and some unexpected and interesting insights. The text were generally well written, except for some issues listed below, and the figures were made nicely.

Major issue:

There was no explanation/justification regarding why the sample number for autosomal markers-based analysis is 123, much smaller than that for the Y chromosome markers-based analysis (356). It’s unknown whether the latter is inclusive of the former. One could easily argue that the lower diversity seen for the autosomal gene pool than that of the Y chromosome gene pool is due to the smaller sample size for the former. In fact, the number of samples were not provided anywhere in the main text other than the abstract. While some observations are novel, their biological significance is unclear or explained.

Writing issues: 

1. Abstract: considering adding a phrase before describing the results to indicate the statements to follow were based on the results.

2. I find the introductionsection a bit lengthy and confusing, failing to provide a clear and strong statement of the research gaps and the rationale for the study.

Insufficient clarity and description:

1. 171: was DNA extracted from blood and saliva for samples or some from blood and some from saliva?

2. The information for the number of samples is missing and lack any explanation/jusfitification

3. L216: the use of GeneGeo software was mentioned but without any references or website. Is it still a private tool? The same question goes for the Y-base database. It would useful be to make the utility and resource available.

4. I couldn’t find the caption for the supplementary figures.

Typographical errors and others: 

1. In many instances, a comma was used for the decimal point of the values, e.g.,“~3,5 ky” in L57, and “2,5 kya” in L63 and L75; the values in Figure 2 and Figure S3. Additional errors can be seen in L31, 

2. an extra space between Y and -chromosome.

3. Figure 6: the labels of “A” and “B” are missing from the figure.

4. For dating, authors used the CE and BC/AD systems. It would be less confusing if staying with one of the systems.

5. Grammatical issue for the sentence from L654-655 (requires to the use of a clause for the verb "analyze").

Author Response

Dear Reviewer,

Thank you very much for your detailed consideration of our research, your comments and suggestions for improving the article.

We tried to take them into account, more details are in the answers below and in the modified manuscript file.

Major issue:

Thank you for your comment! In the Materials and Methods section, we have added an explanation of the differences in the sizes of the population samples for the Y-chromosomal and genome-wide arrays associated with different numbers of markers (80 Y-SNP and ~4,000,000 autosomal SNP). In Supplementary Table S2, we have added the sizes of samples studied for both autosomal and Y-chromosomal markers (N=112) and only for autosomal SNP array (N=11). In the abstract, we have also clarified the distribution of samples by groups (Finnic-speaking peoples and Russian populations for comparison for both genetic marker systems).

Writing issues: 

Thank you! Together with our colleagues who specialize in history and linguistics, we have revised and restructured the texts of the annotation, and partially the introduction and conclusion.

Insufficient clarity and description:

1.  Thank you, we have clarived it in Methods (some from blood and some from saliva).

2.  Thank you, we have added an explanation in the Materials and Methods section.

3.  Undoubtedly, it would useful be to make the GeneGeo software and Y-base database available, but there is no such possibility in the near future.

4.  Captions for Supplementary Figures and Tables are given after the "Conclusion" section in “Supplementary Materials”.

Typographical errors and others: 

 Thank you, we tried to fix it.

Best regards,

Anastasia Agdzhoyan and co-authors

Reviewer 2 Report

Comments and Suggestions for Authors

The paper explores the genetic characteristics of the Baltic Finnic populations in Russia using the Y chromosome and genome-wide autosomes. The authors describe the genetic diversity among these populations, showing that the Y gene pool is more diverse than the autosomal one. Using a large sample of 357 individuals from 8 Baltic Finnic populations, the study divides the gene pool of these populations into different genetic components, with a prevalence of certain components among specific groups. Furthermore, the evaluation of the phylogeography of the N3a4-Z1927 subhaplogroup suggests that these populations share a common ancestor with Northern and Central Russian populations dating back to the Roman Iron Age.

The paper is innovative in that it is the first study to provide a comprehensive and detailed view of the gene pool of Baltic Finnic populations of Russia using two highly informative genetic marker systems: Y-chromosome markers and genome-wide autosomal data. Previous literature has been limited to fragmented or group-focused studies, without a broad and detailed coverage of the populations involved. The combined approach of statistical, bioinformatic and cartographic techniques offers a new perspective on how these groups were genetically formed and their relationship with the neighbouring populations.

The article is well-structured and follows a logical sequence:

·         The introduction is well-written. It provides a comprehensive historical and geographical context of the Baltic Finnic populations. It explains the significance of migration and genetic influences from Siberia and Eastern Europe. It effectively links to the need for an in-depth genetic study of these understudied populations.

·         The materials and methods section is detailed and accurately describes the sample collection and techniques used. The methodologies for statistical and bioinformatic analysis are explained rigorously, with references to the software used for the analyses (e.g., GenomeStudio, Statistica, PLINK).

·         The results are presented clearly, with explanatory figures that help to understand the distribution of genetic components. Although some figures may be difficult to interpret for non-expert readers, the explanations present in the text are detailed and facilitate the interpretation of the results, which is essential due to the complexity of the topic covered.

·         In the discussion section, the authors compare the collected data with previous studies and provide a clear explanation of the observed divergences between populations. The conclusions are on point with the findings and highlight the need for further studies.

Overall, the paper represents a contribution to the knowledge of the structure and application of genetic markers for population studies. This study opens to future research on less-studied populations and offers ideas for further genetic and historical investigations.

Author Response

Dear reviewer,

Thank you very much for reviewing our article and highly appreciating the work done.

Best regards,

Anastasia Agdzhoyan and co-authors